# Developing contents for a digital adherence tool: A formative mixed-methods study among children and adolescents living with HIV in Tanzania

Iraseni Ufoo Swai[1,2]*, Lisa Lynn ten Bergen[3,4], Alan Mtenga[1,5], Rehema Maro[1,5], Kennedy Ngowi[1,2], Benson Mtesha[1], Naomi Lekashingo[1], Takondwa Msosa[2,6], Tobias F. Rinke de Wit[3,7], Rob Aarnoutse[8], Marion Sumari-de Boer[1,3,9]

1 Kilimanjaro Clinical Research Institute, Moshi, Tanzania, 2 UMC Amsterdam, Location AMC, Amsterdam, the Netherlands, 3 Amsterdam UMC, location University of Amsterdam, Department of Global Health, Amsterdam Institute for Global Health and Development, Amsterdam, the Netherlands, 4 Vrije Universiteit Amsterdam, Amsterdam, the Netherlands, 5 Ifakara Health Institute, Bagamoyo, Tanzania, 6 Helse Nord Tuberculosis Initiative, Kamuzu University of Health Sciences, Blantyre, Malawi, 7 PharmAccess Foundation, Amsterdam, the Netherlands, 8 Radboud university medical center, Department of Pharmacy, Research Institute for Medical Innovation, Nijmegen, the Netherlands, 9 Institute of Public Health, Kilimanjaro Christian Medical University College, Moshi, Tanzania

* i.swai@kcri.ac.tz

**Data Availability Statement:** Data cannot be made public as the regulations in Tanzania do not allow for data sharing without arranging a data transfer

## Abstract

Optimal adherence (>95%) to antiretroviral treatment (ART) remains a challenge among children and adolescents living with HIV (CALHIV). Digital adherence tools (DAT) with reminder cues have proven feasible among adult people living with HIV (PLHIV), with some concerns about the risk of HIV status disclosure. We aimed to assess the needs, contents and acceptability of an SMS-based DAT among CALHIV. We first conducted a survey to understand potential barriers to using DAT among CALHIV, then tested the DAT intervention among purposively selected participants. The DAT intervention included using the Wisepill device, receiving daily reminder SMS and receiving adherence reports on how they had taken medication in the past month. The content of the reminder SMS differed over time from asking if the medication was taken to a more neutral SMS like "take care". Afterwards, we conducted exit interviews, in-depth interviews, and focus-group discussions. We analysed quantitative findings descriptively and used thematic content analysis for qualitative data. We included 142 children and 142 adolescents in the survey, and 20 of each used the intervention. Eighty-five percent (121/142) of surveyed participants indicated they would like to receive reminder SMS. Most of them (97/121-80%) of children and 94/121(78%) of adolescents would prefer to receive daily reminders. Participants who used the DAT mentioned to be happy to use the device. Ninety percent of them had good experience with receiving reminders and agreed that the SMS made them take medication. However, 25% experienced network problems. Participants preferred neutral reminder SMSs that did not mention the word 'medication', but preserved confidentiality. The provided adherence reports inspired participants to improve their adherence. None of the participants experienced unwanted disclosure or stigmatisation due to DAT. However, 5% of adolescents were

agreement with the receiver of the data. The authorities will have to approve the data-sharing request by approving a data transfer agreement. Currently, Tanzania is developing their own data repository where meta-data of the study will be shared soon. To gain access to the data the receiver should contact the KCRI administrator (kcriadmin@kcri.ac.tz).

**Funding:** The whole project including human resources was funded through one source, which is the European and Developing Countries Clinical Trials Partnership (EDCTP) under the senior fellowship plus TMA2818. The funders had no role in study design, data collection and analysis, decision to publish, or preparation of the manuscript. None of the authors received direct salary from the funder.

**Competing interests:** The authors have declared that no competing interests exist.

concerned about being monitored daily. This study showed that DAT is acceptable and provided insight of the needed SMS content for a customized DAT for CALHIV.

## Author summary

CALHIV are required to take antiretroviral medication on time, every day, for the rest of their lives. This is necessary to suppress the virus and live a healthy life. Maintaining that consistency is not easy. Digital tools that remind medication time have proven feasible among adult PLHIV. However, there are concerns about HIV status disclosure due to the contents used in the SMSs. We tested a DAT intervention in which participants used the Wisepill device to store medication, with which they received daily reminder SMS 30 minutes before medication intake time. The device recorded every lid opening as medication intake and created an adherence report in the form of a graph that was presented to participants after completing one month. We sent SMS content that differed over time from asking if the medication was taken to more neutral SMS like "take care". After one month, we asked participants their opinions about the intervention. Most participants were happy to use the device and to receive neutral SMS content that did not mention 'medication' and which preserved their confidentiality. Provided adherence feedback motivated participants to improve their adherence. However, some participants experienced network challenges, and 5% of adolescents were concerned about being monitored daily. This study provided an understanding of the needed content for SMS-based medication reminders.

## Introduction

Children and adolescents living with HIV (CALHIV) form a significant part of people living with HIV (PLHIV). Globally, it was estimated that 1.7 million children aged 0–14 years were living with the human immunodeficiency virus (HIV) in 2021, and 88% resided in Sub-Saharan Africa (SSA) [1,2]. Tanzania was mentioned among the top six countries in SSA with the highest number of CALHIV aged 0–19 years. Tanzania had 96,000 children aged 0–14 years living with HIV and only 60% of them were on antiretroviral therapy (ART) [1,3].

The UNAIDS' 95% fast-track treatment goals aim to end the HIV epidemic by 2030 by having 95% of PLHIV know their status, 95% of those who know should have initiated ART, and 95% of those on ART to have suppressed viral loads [4]. The introduction of combination ART has significantly changed the face of HIV/AIDS by reducing morbidity and mortality rates. Good adherence to ART is needed to achieve the last 95% goal of sufficient viral suppression [5–8]. Adherence to treatment poses a significant challenge for people living with HIV since treatment is for life.

Several studies have shown that CALHIV are less likely to have a suppressed viral load ($\leq 200$ copies/ml) compared to adults due to poor medication adherence ($< 95\%$) and poor retention in care [9–12]. A recent study done in Mwanza, Tanzania, reported that only 62.5% and 65.3% of children and adolescents, respectively, reached optimal adherence ($> 95\%$) by pharmacy refill [13]. In another study in Dar es Salaam, 37% of adolescents could not reach optimal adherence [14]. Because HIV infection requires lifelong treatment, CALHIV face more years of needed optimal adherence than adults. Moreover, children depend primarily on their parents or caregivers regarding disease management.

Multiple barriers to adherence have been reported by CALHIV. Among adolescents, the mentioned barriers include stigma, adverse effects of ART, lack of assistance, depression and

forgetfulness [15–18]. Reported factors among caregivers for CALHIV are forgetfulness, long clinic waiting hours, busy schedules and limited knowledge about HIV [19,20]. Children and adolescents in boarding schools face particular adherence barriers like fear of status disclosure mainly due to drug packaging, lack of privacy during medication intake time, challenges with drug storage and lack of a structured support system in schools [17,18,21]. Therefore, developing interventions to overcome adherence barriers is of utmost importance to maintain optimal viral suppression.

In recent years, interest has grown in using digital adherence tools (DAT) with reminder cues through short message services (SMS) for improving adherence to treatment and retention in care [22–27]. Several studies have found DATs with reminder SMS feasible among adult PLHIV [28–31]. However, there is limited evidence of its effectiveness among CALHIV in improving adherence [32–34]. Also, there are concerns about unwanted disclosure of the HIV status due to the wording used in the SMS texts [35–40].

As Tanzania has high penetration of mobile phones (89%), we believe that phone-based interventions can potentially assist CALHIV with taking their medication [41]. Through DAT interventions, healthcare workers can monitor medication adherence with access provided through online adherence reports and subsequently provide personalized adherence feedback to the individual. Hence, whereas the DAT device addresses the barrier of 'forgetfulness' by sending SMS reminders, personalized feedback using adherence reports during clinic visits can address specific barriers faced by CALHIV.

This study aimed to (1) assess the needs and contents for an SMS based DAT and (2) assess the acceptability of the DAT intervention.

## Methods

### Study design and study population

In this formative study, we used a convergent parallel mixed-methods design. We collected quantitative and qualitative data concurrently and analysed them independently to answer our study objectives [42]. The study consisted of a survey and a one-month DAT intervention in a selection of participants, followed by a semi-structured exit interview, in-depth interviews (IDIs) and focus-group discussions (FGDs) among those who had received the DAT intervention. The detailed study design, including the sample size calculation, has been described elsewhere [43]. The study was approved by the College Research and Ethical Review Committee (CRERC) of Kilimanjaro Christian Medical University College (KCMUCo) and the National Health Research Ethics Sub-Committee (NatHREC) of the National Medical Research Institute (NIMR) of Tanzania.

We conducted the study in Kilimanjaro Region, Tanzania. Children (aged 0–14 years) with their caretakers and adolescents (15–19 years) living with HIV were eligible if they met the following inclusion criteria: (1) attending Care and Treatment Centres (CTC) in the recruiting health facilities, (2) willing to use the DAT and receive SMS, (3) having received ART for at least six months, and (4) having a mobile phone with a registered SIM card (for children this was the caregiver). Exclusion criteria were admission to a hospital at study entry or previous or concurrent participation in other digital adherence trials. We provided phones for those lacking one.

### Study procedures

Fig 1 below summarises the study procedures. We used an extensive approach with guardians/parents to consider whether the HIV status was disclosed to the child and asked for assent from children older than eight years who were aware of their status. For participants <18

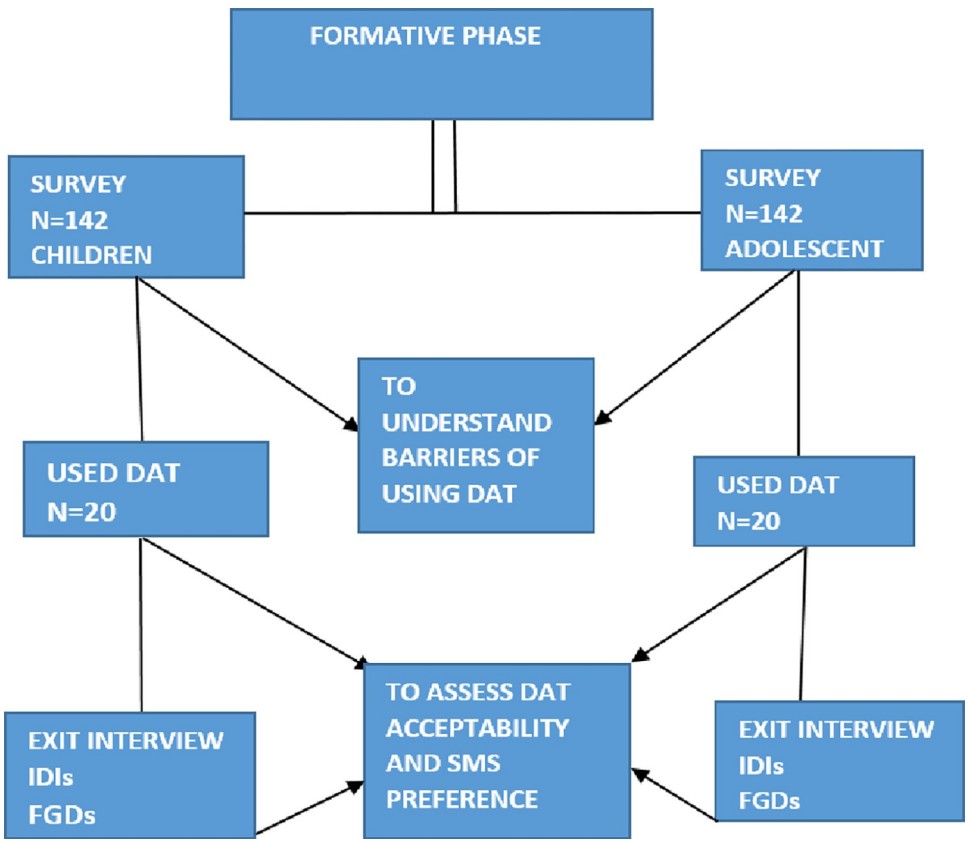

**Fig 1. The design of the overall study flow.**

years old, we obtained written informed consent from their caregivers or parents. We obtained written informed consent from adolescents aged 18 to 19 years. We conducted a second written informed consent procedure for participants enrolled in the DAT intervention.

## Survey

We conducted a survey to understand potential barriers of using DAT among CALHIV. We used convenience sampling to approach 142 children and 142 adolescents. Using semi-structured questionnaires, we recorded demographic characteristics, disease and treatment information, usual medication intake time, the need for SMS reminders, and suggested frequency and timing of the SMS. We collected data on the participant's adherence to ART medications, i.e. the date and number of dispensed, swallowed and missed pills. This was used to calculate adherence by the self-report and pharmacy refill methods. We also recorded information about owning and using mobile phones, experience with communicating through SMS, network issues and other phone-communication challenges.

After the survey, selected participants were provided with the Wisepill devices and enrolled in the DAT intervention.

## The Digital adherence tool (DAT) intervention

We applied purposive sampling when selecting participants for the DAT intervention. We purposively selected participants to get heterogeneity in age, adherence levels and levels of health facilities. Participants used our DAT for one month to understand the device's

mechanism and receive different reminder messages. For children, SMSs were sent to their parents/caregivers, as well as for adolescents without personal phones.

We used the Wisepill RT2000 pillbox for the DAT intervention, an internet-enabled medication dispenser. The device registers each time it is opened and automatically generates a report indicating the time of intake per day. When the dispenser is opened, it sends an electronic medication record to a central management system (Wisepill Web Server) through the GPRS data network. If patients forget to take their medication at the prescribed time, the system sends a text message to their mobile phones. We sent reminder SMSs, firstly half an hour before intake time and secondly, one hour after the usual time of intake if the participant forgot to open the pillbox at the prescribed intake time. In the first week, we used general SMS content, like: *"Don't forget to take your medication!"* We adopted this content from our previous study. (29) In the second, third, and fourth weeks, more neutral content was used, such as *"Remember your health", "The time is near"*, or *"Remember!"*

The research team monitored the patients' medication events through online authorized access, displaying a graph report regarding the medication intake of patients. The nurse discussed this report with participants during adherence feedback sessions at the end of the intervention. The trans-theoretical model (TTM) of behavioral change was applied to shape the intervention as practical guidance during the feedback consultation [44]. The model was modified to fit both children and adolescents. Children who were aware of their HIV status joined their parents/caregivers during the feedback section.

## Exit interviews, In-depth interviews (IDIs) and Focused Group Discussions (FGDs)

Following one month of using the intervention, we conducted exit interviews, IDIs and FGDs to understand participant's opinions about the appearance and functionality of the device, their preference for the received SMS contents and their perspective on the provided adherence feedback. We also collected information on the potential of using educational or motivational SMS. We collected data in Swahili by trained research assistants not part of the participants' standard HIV care. We used semi-structured questionnaires for the exit interviews and topic guides for the IDIs and FGDs. We also included in the IDI and FGDs children older than eight years whose HIV status had been disclosed. A research assistant led the FGDs with different groups of children, adolescents, and caregivers. The FGD was conducted by the research team (interviewer, facilitator, and reporter) with participants (6–8 in each FGD). The interviewer used an interview guide in an interactive process. We then transcribed interviews verbatim and then translated them into English.

## Theoretical framework and data analysis

To answer the study objectives, we computed descriptive frequencies of variables to give an overview of the data from the survey and exit interviews. We analyzed data using SPSS Statistics 27. For data that were not normally distributed, we reported outcomes as medians and the interquartile range (IQR) to indicate the measure of dispersion.

For qualitative data, we conducted a thematic framework analysis. We applied the Theoretical Framework of Acceptability (TFA) post-intervention [45]. The TFA is defined as a multifaceted framework that reflects the extent to which people delivering or receiving a health care intervention consider it appropriate, based on anticipated or experiential cognitive and emotional responses to the intervention. We applied all seven TFA constructs, which are (1) *Affective attitude*, whether CALHIV were satisfied with the DAT intervention; (2) *Intervention coherence*, the extent to which CALHIV understood the DAT and how it worked; (3) *Perceived*

*effectiveness*, the extent to which the DAT intervention increased CALHIV's motivation to be adherent to ART; (4) Intervention *burden*, the amount of effort that was required to participate in the DAT intervention; (5) *Ethics* the extent to which the DAT had a good fit with an individual's value system, such as religion, beliefs, rights and culture; (6) *Opportunity costs*, the benefits, profits or values that were given up to engage in the DAT intervention; (7) *Self-efficacy*, the CALHIV's confidence that they can perform the behavior required to participate in the DAT.

We applied inductive and deductive coding. We first summarized transcripts to get familiar with the data. IS, LB and AM created memos based on the first six interviews, and we developed a preliminary codebook. We organized data inductively into each TFA construct. We discussed the codebook and then applied codes to the remaining transcripts. We used NVivo version 12 pro for data organization.

## Results

### Number and characteristics of participants in the sub-studies

A total of 284 (142 adolescents and 142 children and their caregivers) participated in the survey (Table 1). We conducted 48 IDIs in total: 8 with children older than eight years to whom their HIV status had been disclosed, 20 with adolescents and 20 with caregivers of the children (see S1 Appendix & S2 Appendix for the characteristics of each). We conducted five FGDs: two with adolescents, two with caregivers and one with children whose HIV status had been disclosed.

### Acceptability of DAT

**Exit interviews, IDI and FGD.**   We present the results about the preferred SMS contents and the acceptability of the intervention according to the seven constructs of the TFA. Table 2 describes the participants' general experiences of using the DAT.

### Affective attitude

Most participants indicated having had a good or very good experience receiving reminder SMSs (Table 2). Participants liked the device, its appearance, shape, and colour, mainly because it was difficult for others to know its function. It was a helpful tool to safely store medication and preserve confidentiality, especially among participants attending boarding schools, where pupils spend their entire time on campus. Parents/caregivers of adolescents expressed happiness that the device had a monitoring feature to track whether adolescents were taking their medication.

Moreover, participants shared positive feelings about the adherence feedback (S5 Appendix). All caregivers and 75% of adolescents positively evaluated the graphs' content. Participants indicated that it provided a clear overview with different colours about daily medication intake. Nevertheless, 25% of adolescents were frustrated with the feedback because it showed that they did not open the box, often due to technical malfunctioning, while they were ingesting pills.

Most participants liked the SMSs that used general language, did not mention the word 'medication' and were not easy for others to understand the purpose of the SMS (S3 Appendix & S4 Appendix). Among adolescents, the most preferred SMS contents were '*Your health is important*' (90%), '*The time is at hand*' (95%) and '*Care for your health*' (95%). Caregivers of children frequently preferred '*Do not stop protecting the child*' (80%), '*Care for the child's health*' (85%), and '*The time for the child to use is at hand*'. (90%).

**Table 1. Survey participant's characteristics and needs for reminders.**

| Variable | Children (N = 142) | Adolescents (N = 142) |
|---|---|---|
| | Frequency (%) IQR | Frequency (%) IQR |
| **Sex** | | |
| Male | 68 (48%) | 77 (53%) |
| Female | 74 (52%) | 65 (47%) |
| **Age** | 9(IQR 7–12) | 18(IQR 17–18) |
| **Level of education** | | |
| None | 16 (11%) | - |
| Primary | 124 (87%) | 32 (23%) |
| Secondary | 2 (1%) | 102 (72%) |
| Tertiary | - | 8 (6%) |
| **HIV status disclosed** | | |
| Yes | 52 (37%) | 141 (99%) |
| No | 90 (63%) | 1 (1%) |
| **HIV status** | | |
| Asymptomatic | 138 (97%) | 137 (96%) |
| Symptomatic | 2 (1.4%) | 4 (3%) |
| AIDs converted | 2 (1.4%) | 1 (0.7%) |
| **Own or use phone*** | | |
| Yes | 133 (94%) | 105 (74%) |
| No | 9 (6%) | 37 (26%) |
| **If Yes, Sharing phone with others*** | | |
| Yes | 20 (15%) | 24 (23%) |
| No | 113 (85%) | 81 (77%) |
| **SMS experience** | | |
| Sending | 105(78%) | 111(93%) |
| Reading | 129(96%) | 119(99%) |
| Receiving | 131(98%) | 118(98%) |
| **Ever experienced network problem** | | |
| Yes | 42(30%) | 38(27%) |
| No | 93(66%) | 82(58%) |
| **Always takes medication on time** | | |
| Yes | 136(96%) | 112(79%) |
| No | 6 (4.2%) | 30(21%) |
| **If No, Would like to be reminded** | | |
| Yes | 5(83%) | 28(93%) |
| No | 1(17%) | 2(7%) |
| **Anyone assisted in reminding to take the medication** | | |
| Yes | 94(66%) | 108(76%) |
| No | 48(34%) | 34(24%) |
| **Like to receive reminder SMS** | | |
| Yes | 121(85%) | 121(85%) |
| No | 21(15%) | 21(15%) |
| **If yes, how often receive SMS** | | |
| Daily | 97(80%) | 94(78%) |
| Weekly | 5(4%) | 18(15%) |

(*Continued*)

**Table 1.** (Continued)

| Variable | Children (N = 142) | Adolescents (N = 142) |
|---|---|---|
| | Frequency (%) IQR | Frequency (%) IQR |
| Monthly | 19(16%) | 9(7%) |

*For children, we asked whether the caregiver owns or uses a phone

IQR = Interquartile range

Participants disliked receiving the SMS mentioning the word 'medication'. The specific content of the first SMS, '*your time to take medication is near; you are reminded to take your medicines as directed by healthcare workers*, was not preferred by 8 (40%) caregivers and 6 (30%) adolescents. The SMS was too long, contained too much information and would raise questions when seen by others.

Adolescents had several opinions about SMS. Some suggested that an abbreviation would have been sufficient to make them remember. Some of them were irritated by the content of the SMSs and doubted the overall effect of the reminder SMS; for them, SMSs just increased the tension of unwanted disclosure of status.

Participants also suggested educational topics to be sent in SMS, including how to prevent HIV transmission to others, information about nutrition, entrepreneurship, reproductive health, and family planning methods. Caregivers wanted education on how they can best disclose to their children their HIV status.

## Intervention coherence

Participants understood how the device worked. Most participants reported that the device was functioning well and always received the first reminder SMS in time, and if they opened the device, they did not get the second reminder SMS. Participants found the device easy to open and pack medication. Nevertheless, some participants thought it had cameras so healthcare workers could see them.

However, we also found that adolescents often faced difficulties using the Wisepill box appropriately, affecting the results of their adherence levels. A minority of adolescents reported a low battery and having no time to change the device, having lost their mobile phone or forgetting to bring it during travel. Specifically, in one case, because of a desire to discard the medical pill bottle, an adolescent put all medications in the pillbox, resulting in a situation where the box could not close and communicate properly.

Generally, the reminder SMSs were found to be easy to understand. Adolescents said that since they were told at study entry that they would receive reminder SMS, they were expecting them. When they received the SMS from the study number, they automatically understood that it was for reminder purposes.

All adolescents and children's caregivers indicated that they had seen and understood the graph during the feedback consultation and believed that the nurses explained it clearly. They knew how the device communicated about their adherence and how that information appeared in the graph.

## Perceived effectiveness

Participants considered daily SMS effective. Also, participants mentioned that SMS timing of 30 minutes before medication time was reasonable since it allowed them some preparation time to plan and reach home on time. Nevertheless, others suggested reducing the time to 20

**Table 2. Participant's experience with using DAT.**

| TFA CONSTRUCT | VARIABLE | CHILDREN N = 20 Frequency (%) | ADOLESCENTS N = 20 Frequency (%) | EXAMPLE OF QUOTES |
|---|---|---|---|---|
| *Affective attitude* | **Experience with receiving reminder SMS** | | | "When I was included in this project, I was really happy because I thought it would help me to know when my child has taken his medications and when he hasn't. So, I know what is causing his disease progress to be bad or good sometimes." (IDI, caregiver) "Yes, I liked it much better than the hospital container. I can go with it anywhere; no one notices I carry medication". (IDI, adolescent) "I didn't like it because there was a time when I was taking medication. However, they [the pills] were kept in another pill bottle than the Pillbox device, so when I came for a clinic visit, I found that my adherence levels had decreased."(IDI, adolescent) |
| | Very good | 10(50%) | 3(15%) | |
| | Good | 8(40%) | 15(75%) | |
| | Neutral | 1(5%) | 2(10%) | |
| | Not good | 1(5%) | 0(0%) | |
| | **Think of Wisepill box** | | | |
| | It is a very nice | 7(35%) | 7(35%) | |
| | It is nice | 12(60%) | 13(65%) | |
| | Neutral | 1(5%) | 0(0%) | |
| | **Liked the contents of the feedback** | | | |
| | Yes | 20(100%) | 15(75%) | |
| | No | 0(0%) | 5(25%) | |
| *Intervention coherence* | **Understands the graph** | | | "My grandchild would often say: 'Grandma, the message just came in. Where is the device so I take my medication? They will see us if I do not take them on time.' I had told her that if she doesn't take it on time, the device takes her picture" (FGD, caregiver) "The graph showed that there was a day that I didn't take my medication at all and there was a day that I took my medication too late." (IDI, child) |
| | Yes | 18(90%) | 20(100%) | |
| | No | 1(5%) | 0(0%) | |
| *Perceived effectiveness* | **Confirms that SMS makes you take medication** | | | "In the past, if you were tired, you would say: 'I shall take medication tomorrow', but now you know you are being monitored. You just have to take." (IDI, adolescent) "My child used to refuse to take the medication, but now when he hears the text messages, he says: 'That is my message' and goes to take the device, so that I can give him the medication." (FGD, caregiver) "I had my problems and not the matter of forgetting. I was just ignoring it when messages came in. I pretended that I was busy and I didn't take the medication." (IDI, adolescent) |
| | Yes, always | 16(80%) | 15(75%) | |
| | Most times | 2(10%) | 5(25%) | |
| | Mostly not | 2(10%) | 0(0%) | |
| | **Thinks feedback graph can improve adherence** | | | |
| | Yes | 18(90%) | 19(95%) | |
| | No | 2(10%) | 1(5%) | |
| *Burden* | **Experienced difficulties using the device** | | | "I had a bad experience because you have taken your medication and then they accuse you that you haven't taken your medication, and the other message comes." "Anxiety existed within me [about being monitored]. However, I knew it was good because the device was reminding me." (IDI, adolescent) "Ever since I received this device, everyday it's 'you have not given the child medication, you have not given medication', while I never skip." (IDI, caregiver) |
| | **Yes** | 1(5%) | 1(5%) | |
| | **No** | 19(95%) | 19(95%) | |
| | **Experienced a mobile network problem** | | | |
| | Yes | 5(25%) | 4(20%) | |
| | No | 15(75%) | 16(80%) | |
| *Ethics* | **Concerned about being monitored daily** | | | "I have not seen any disadvantage of the device. . ... Just one! If you do not use it, you will be detected that you did not take your medication." (IDI, child) "Only my family saw the SMS, and they find it normal since they know my status. But no one else outside the home saw them. I was hiding them from my friends, friends are not people to trust; they may see it and make you the headline." (IDI, adolescent) |
| | Yes | 0(0%) | 1(5%) | |
| | No | 20(100%) | 19(95%) | |
| | **Using DAT lead to unwanted disclosure of HIV status** | | | |
| | Yes | 0(0%) | 0(0%) | |
| | No | 20(100%) | 20(100%) | |

(*Continued*)

**Table 2.** (Continued)

| TFA CONSTRUCT | VARIABLE | CHILDREN N = 20 Frequency (%) | ADOLESCENTS N = 20 Frequency (%) | EXAMPLE OF QUOTES |
|---|---|---|---|---|
| *Opportunity cost* | **Incurred personal costs to participate in DAT** | | | "To be very frank, the place I am staying, to ask permission every time while I do not want people to know my condition. I don't like that. So, the device should be adjusted to be stable, so people are not called to the clinic every time it has problems." |
| | Yes | 0(0%) | 0(0%) | "I see the cost is that you should have a phone since the device doesn't ring an alarm to say it will remind you." (FGD, caregiver) |
| | No | 20(100%) | 20(100%) | |

or even 5 minutes closer to intake time to prevent forgetting to take medication despite receiving the reminder SMS.

The device was considered effective in making participants take their medication on time. This was through reminders that addressed forgetfulness problems, but also the feeling that they were being monitored stimulated them to adhere to pill intake time. Some participants indicated being worried about living without the Wisepill box. None of the participants reported ever opening the device to satisfy the researchers or to avoid getting the second reminder SMS.

Even though the device was perceived as effective in reminding medication time, other factors, i.e. a non-confidential environment, the phone being in silent mode, the phone not being charged, or network problems still hindered adherence to treatment among participants. Sometimes, participants ignored the SMS, thinking they would take the medications later.

Nearly all participants indicated that receiving feedback on their adherence motivated them to improve adherence to their medication in the next month. Respondents claimed that the information in the graph increased their motivation and confidence to take medication every day and reach the level of adherence as agreed with the nurse. Interestingly, a few participants mentioned that the device encouraged greater accuracy in discussing their adherence behaviour with the nurse.

## Intervention burden

Most participants deleted the reminder SMS immediately after reading them, regardless of whether they liked the content or not. The reasons for deleting were to empty phone space for a few participants, but most participants did that to prevent unwanted disclosure.

The fact that the DAT sends reminders through a mobile phone created a challenge for participants without a personal phone. This was more noticeable among participants in boarding schools, as they are not allowed to have phones at school. In this case, the reminder SMS was received by a school caretaker or a parent/caregiver at home who would then call or forward the SMS to the school's caregiver. Some wished the device could ring an alarm during medication time instead of sending an SMS through the phone.

Twenty-five percent of caregivers and 20% of adolescents experienced mobile or device network problems (Table 2). Consequently, they would receive the second reminder while they had already opened the device. Some changed their medication intake time without informing the research team, resulting in inadequate SMS traffic.

None of the interviewed participants found the feedback consultation too long or that coming to the clinic was a burden to them. We asked participants about being monitored daily, and even though some felt anxious, most individuals did not share personal concerns about adherence counselling using real-time feedback.

## Ethics

Participants considered the device to be user-friendly. None of the participants faced stigmatisation because of using the device. None of the participants reported the occurrence of unwanted HIV status disclosure. Some explained that, since the SMS were general and did not mention personal names, even if others happened to see the messages, they could explain that it was not meant for them.

## Opportunity costs

None of the participants incurred additional personal costs, such as costs related to transporting expenditures or time lost from work to participate in this intervention. The feedback session aligned with their regular appointment visits. Participants only had to return to the clinic in the case of technical device issues and when we asked participants to come to change their devices.

# Discussion

This formative study aimed to assess the needs, contents, and acceptability of the DAT with reminder cues and tailored feedback on adherence among CALHIV. We found that CALHIV need DATs, and they find it highly acceptable. Most participants had a positive experience using the device and preferred neutral SMS content that did not mention the word 'medication'. The provided feedback reports on adherence inspired participants to improve their adherence.

## Need for reminders

Results from the survey indicate that children's caregivers and adolescents who had no people to remind them about the medication time needed reminders. They were interested in receiving daily reminder SMS before the usual medication time or during the evening. Participants in the DAT intervention confirmed this. Similar results were found in a study done in Peru, where participants preferred text messages over recorded voice messages or phone calls. [38] Participants found text messages easier, readily accessible, and more confidential, although they suggested changing the SMS content daily or weekly to avoid boredom. In another study in Uganda, scheduled daily messages were preferred as they reduced the chance of missing doses compared to weekly or triggered SMS [46].However, some participants in that qualitative study were worried that frequent text messages for each dose would lead to phone dependency for medication intake, causing more problems in case of phone malfunction [47]. Furthermore, a meta-analysis of randomized control trials reported that scheduled daily text message reminders improved adherence outcomes [48].

## SMS contents

Preference of SMS content depended on whether participants had disclosed their HIV status to people around them when receiving the SMS and whether they shared their phones. This was also reported by studies conducted in Kenya and Uganda, in which open contents SMS were found frustrating and increased the risk of unwanted disclosure of HIV status [46,47]. In the Kenyan study among postpartum women, SMS content preference was also related to the status of HIV disclosure to the partner/husband. Women whose partners were unaware of their HIV status preferred SMS content with low verbal immediacy [47]. Some suggested using code words or setting passwords, but the latter depends on the type of phone owned [38,49]. The ability of the SMS to preserve confidentiality and privacy was the most critical

factor discussed by participants for the acceptability of the SMS-based mobile health intervention.

Similar to our previous acceptability studies among PLHIV, participants liked the SMS contents that were motivating in nature because they made them feel seen and cared for [30,46,47].

In this pilot study among CALHIV, some adolescents were irritated by the overall idea of reminder SMSs, while some suggested that just an abbreviation would be enough to remind them. However, this was not observed among the children. These could be due to the fact that more adolescents (23%) shared their mobile phones than caregivers (15%) or because adolescents received the SMS and responded to questions themselves, while for children, it was the parents/caregivers playing the most role.

Also, participants suggested several topics they would like to receive educational messages about, for example, receiving education on how to disclose HIV status to a child. This indicates that participants consider mobile health interventions promising and suitable for addressing HIV-related issues if they preserve confidentiality.

## Acceptability of the intervention

This study found that an SMS-based digital adherence tool with personalized adherence feedback was needed and highly acceptable among CALHIV. Considering *affective attitude*, we found that most participants were satisfied with the intervention. In other studies, participants also described that the device was easy or convenient to use and convenient in preserving confidentiality since it didn't resemble the traditional pill container [27,46,50]. Additionally, our study found that it was a helpful tool to safely store medication and preserve confidentiality among boarding school participants. Unfortunately, we did not record how many participants attended boarding school, this finding came up during qualitative data collection. A study in Kenya among boarding school students reported fear of unintended disclosure due to drug packaging as one of the barriers to ART adherence [17]. Therefore, electronic monitoring devices can potentially solve these adherence challenges.

DAT was *perceived as effective* in reminding medication time and making participants take their medication on time. It was also explained by participants in other studies that DAT trained them to develop good adherence behaviour in taking the pills daily and on time [27,50,51]. Few adolescents were concerned about being monitored daily. However, their caregivers liked the tool's monitoring feature as they could know if their children took medication or not. A qualitative study exploring the perceptions on the effect of electronic monitoring tools on adherence explained that even though electronic monitors created a sense of pressure by forcing participants to adhere, most participants found it beneficial in assisting them in reaching their adherence goals and live a healthy life [52].

Furthermore, we found that most participants understood how the DAT works and how it could improve adherence (*intervention coherence*). In other similar studies exploring the acceptability of DAT devices, participants explained a good understanding of how to open the device, fill it with medication and record missing doses if the device was not opened [46,52]. Nevertheless, a few participants described not using the intervention as intended. They packed more medication that could fit in the device, and some swallowed medicine from the packaging container instead of from the Wisepill box. A feasibility study in Atlanta reported that for 7% of the monitored doses, participants did not open the Wisepill device but swallowed medication from their pockets [34]. Future studies should properly train users on how the intervention works and insist on only taking pills from the device.

All participants received and understood the feedback on adherence graphs as discussed with nurse counsellors. Feedback on adherence motivated them to adhere better to

medication. Participants knew that objective adherence information can improve accurate HIV counselling, where nurses can no longer be misled. Objective monitoring can overcome social desirability bias, a well-recognized bias during patient recall [51]. However, still 25% of adolescents felt dissatisfied with the feedback. This was often due to inaccurate use of the pillbox during the intervention. Some adolescents received lower than-expected percentages but recalled having taken all pills. These contradicting instances may hamper patient-clinician trust that could be perceived as "not being believed". Therefore, caution should be exercised when choosing DAT adherence results as the new gold standard, but rather frame DAT as a complement to patient self-report [53].

Regarding factors affecting acceptability negatively, we found that the intervention could cause a *burden* to participants in case of malfunctioning due to technical or network issues. No signal is sent when there is no network, meaning the system believes no medication was taken, so the second reminder was sent. That created a burden to participants when they kept receiving the second reminder SMS and considered to be accused of not taking their medication while they had. A study conducted in rural Uganda also reported failed data transmission as one of the encountered limitations of the wireless electronic adherence monitors [54]. For longer period interventions, closer and regular follow-up of participants is needed to detect devices that do not communicate and replace them.

We observed other *burden* issues when participants shared phones or did not have access to phones, like students in boarding schools. Similar to what we reported, participants in a pilot study in South Africa recommended that the device should be combined with reminders like alarms or buzzers to assist participants with phone challenges [32].

Nevertheless, SMS-based interventions require participants to have phones, be literate, have access to electricity, and have a good network connection [35,39]. Consequently, these interventions are not always accessible to those who need them. Other reported barriers include battery life, power failures and potential stigmatization due to DAT [29,54,55]. These studies similarly reported that fear of unwanted disclosure of HIV status led participants to delete the SMS shortly after reading them, and some could not take their medications even after reminders due to lack of privacy or psychosocial factors.

Participants suggested receiving educational SMS on different topics, such as the prevention of HIV and how best to disclose the HIV status to a child or a partner. However, this should consider the risk of unwanted disclosure. Hence, careful phrasing of HIV-related health education is essential and should be tailored based on recipient's desires [36,39].

## Study limitations and strengths

Due to the nature of the intervention, only literate participants could participate. Future DAT studies should consider interventions suitable to participants who cannot read, for example, using devices with alarms. A second limitation of this study was the small sample size of participants involved in the DAT study and the duration of one month of using the intervention. Therefore, caution should be taken when generalizing study results.

Also, we used purposive sampling to select participants for the DAT intervention. Attempts were made to approach a diverse set of adolescents. However, it was hard to establish a representative heterogeneity for age. More than half of our participants were 18 years or older. This skewed distribution can be attributed to the fact that informed consent was legally required from parents of underage adolescents. In Tanzania, it is common for young adolescents to visit the clinic alone without their parents, hindering getting consent from their parents. As such, it wasn't easy to recruit participants below 18. We tried to reach the parents by phone

and asked them to visit the clinics to receive more information about the study and, if possible, provide consent for their children to participate.

The study's strengths include that it focused on vulnerable key populations, i.e. CALHIV, while most studies on DAT have been among adults. This study allowed CALHIV to give their opinion and shape a DAT intervention suitable to them and address their unique challenges.

## Conclusion

DAT with reminder SMS and tailored feedback was acceptable among CALHIV. Participants preferred more neutral wording in SMS. We will use preferred SMS in a future clinical trial, which will assess the effectiveness of DAT in improving adherence among CALHIV. Future studies should consider measures to solve network problems, such as increasing the period between the first and second SMS, improving the device to include alarms for participants lacking phones, and improving training about how to use the device properly.

## Supporting information

**S1 Appendix. Characteristics of children in the DAT intervention (N = 20).**
(DOCX)

**S2 Appendix. Characteristics of adolescents in the DAT intervention (N = 20).**
(DOCX)

**S3 Appendix. Adolescents SMS Preference (N = 20).**
(DOCX)

**S4 Appendix. Children SMS Preference (N = 20).**
(DOCX)

**S5 Appendix. Example of an adherence feedback graph.**
(DOCX)

## Acknowledgments

We would like to thank all children and adolescents, together with their caregivers, for taking part in this informative study. Their opinions and ideas shape the future of our interventions to benefit the general society. We thank the healthcare workers in participating facilities and the KCRI REMIND team staff who helped to collect data.

## Author Contributions

**Conceptualization:** Lisa Lynn ten Bergen, Alan Mtenga, Rehema Maro, Kennedy Ngowi, Benson Mtesha, Rob Aarnoutse, Marion Sumari-de Boer.

**Data curation:** Iraseni Ufoo Swai, Benson Mtesha.

**Formal analysis:** Iraseni Ufoo Swai, Benson Mtesha, Naomi Lekashingo.

**Funding acquisition:** Kennedy Ngowi, Marion Sumari-de Boer.

**Investigation:** Iraseni Ufoo Swai, Lisa Lynn ten Bergen, Alan Mtenga, Rehema Maro, Kennedy Ngowi, Benson Mtesha, Naomi Lekashingo, Takondwa Msosa, Tobias F. Rinke de Wit, Rob Aarnoutse, Marion Sumari-de Boer.

**Methodology:** Kennedy Ngowi, Marion Sumari-de Boer.

**Software:** Kennedy Ngowi, Marion Sumari-de Boer.

**Supervision:** Kennedy Ngowi, Tobias F. Rinke de Wit, Rob Aarnoutse, Marion Sumari-de Boer.

**Validation:** Tobias F. Rinke de Wit, Rob Aarnoutse, Marion Sumari-de Boer.

**Visualization:** Iraseni Ufoo Swai.

**Writing – original draft:** Iraseni Ufoo Swai.

**Writing – review & editing:** Lisa Lynn ten Bergen, Alan Mtenga, Rehema Maro, Kennedy Ngowi, Takondwa Msosa, Tobias F. Rinke de Wit, Rob Aarnoutse, Marion Sumari-de Boer.

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
