## [Decision Letter · Decision Letter 0]

30 May 2023

PDIG-D-23-00097

Developing contents for a digital drug adherence tool with reminder cues and personalized feedback: a formative mixed-methods study among children and adolescents living with HIV in Tanzania

PLOS Digital Health

Dear Dr. Swai,

Thank you for submitting your manuscript to PLOS Digital Health. After careful consideration, we feel that it has merit but does not fully meet PLOS Digital Health's publication criteria as it currently stands. Therefore, we invite you to submit a revised version of the manuscript that addresses the points raised during the review process.

Please submit your revised manuscript within 60 days Jul 29 2023 11:59PM. If you will need more time than this to complete your revisions, please reply to this message or contact the journal office at digitalhealth@plos.org. Please include the following items when submitting your revised manuscript:

We look forward to receiving your revised manuscript.

Kind regards,

Haleh Ayatollahi

Section Editor

PLOS Digital Health

Journal Requirements:

2. We do not publish any copyright or trademark symbols that usually accompany proprietary names, eg ©, ®, ™ (e.g. next to drug or reagent names). Please remove all instances of trademark/copyright symbols throughout the text, including ® on pages 2, 8, 9, 10 and 15.

3. We ask that a manuscript source file is provided at Revision. Please upload your manuscript file as a .doc, .docx, .rtf or .tex.

4. In the online submission form, you indicated that "The quantitative and qualitative data used in this study will be made available upon request from the corresponding author". All PLOS journals now require all data underlying the findings described in their manuscript to be freely available to other researchers, either 1. In a public repository, 2. Within the manuscript itself, or 3. Uploaded as supplementary information.

Additional Editor Comments (if provided):

Reviewers' comments:

Reviewer's Responses to Questions

**Comments to the Author**

1. Does this manuscript meet PLOS Digital Health’s publication criteria? Is the manuscript technically sound, and do the data support the conclusions? The manuscript must describe methodologically and ethically rigorous research with conclusions that are appropriately drawn based on the data presented.

Reviewer #1: Yes

Reviewer #2: Partly

2. Has the statistical analysis been performed appropriately and rigorously?

Reviewer #1: Yes

Reviewer #2: I don't know

3. Have the authors made all data underlying the findings in their manuscript fully available (please refer to the Data Availability Statement at the start of the manuscript PDF file)?

Reviewer #1: Yes

Reviewer #2: Yes

4. Is the manuscript presented in an intelligible fashion and written in standard English?

Reviewer #1: Yes

Reviewer #2: No

5. Review Comments to the Author

Reviewer #1: General;

Title: excessive length-difficult to determine primary objective/endpoint and intervention.

Formatting: line numbering would greatly facilitate review by locating content more easily. Therefore, specific comment locations are restricted to section/page references. Paragraph spacing inconsistent throughout. Inclusion of patient quotes not helpful, since uncertain how it was chosen (consistent and representative of others or contrasting).

Abstract: Needs significant revision/reorganization, since it is difficult to follow. See author instructions to assist with organization. Background/objective/population/intervention/endpoint analysis/results/significance would be easier to follow. Survey target (patients are parents/guardian) not identified. Intervention group who “used the intervention” unclear, since DAT intervention not yet described (perhaps a prior SMS-based service???). Unclear until author’s summary provided some additional description of the intervention. Clear explicit statements regarding objectives not provided. Uncertain if “automated adherence reports” was a key inclusion criteria or an endpoint, since objective of the study (adherence vs satisfaction vs barriers vs ???) not identified. Timeline between identification/screening/survey/feedback adjustment not defined. Elements/targets of surveys/interviews not specified. Number of patients included in subsets not identified in results presented. Uncertain if subsets were selected prior to study conduct for analysis. “Patients were happy to use the device” not helpful since device and quantitative response not provided. Unclear how many patients used a device that was normally accessible to multiple individuals beyond the patient and/or parents that would have impacted desirability for neutral messaging. “adherence reports inspired good adherence” not helpful since device and quantitative response not provided. Unclear how focus groups data was collected (structured interviews vs themed response assessed by multiple observers vs ????). Unclear how ARV regimen complexity was accounted for in the responses. Reference to Wisepill device not helpful since no description of the technology is provided and description significantly delayed in the manuscript body. Method for adherence assessment and report format/content unclear. Method/timeline to vary messaging content not described. Method of data distillation from interviews not provided. Unclear at what age the children were able to provide responses in place of/in addition to the caregivers.

Author summary: unclear what functionality is provided by Wisepill and what was additional intervention by the study team. Content and format of reminder messages unclear from present description (“how they took medication over a month” is insufficient). Schedule of message content/intensity not described (except “over time”). Intent to relate adherence feedback to adherence behavior not described in the methods, nor are the data provided to support the statement “Adherence feedbacks motivated good adherence behavior”.

Sample size-unclear which of the objectives/study questions/endpoints were utilized, and no hypothesis testing described that would require such determination. Other than convenience sampling, uncertain how subgroup sizes were determined. Data summary as percentages with small sample sizes (n=20) might be misleading.

Narrative data results generally provided in tables, so can be abbreviated (and potentially combined with discussion) to reduce length. Table data should consider also providing summary statistics for the combined (children, adolescent) columns. 

Strengths/limitations-reconsider organization, since both are mixed within narrative instead of ordered.

General Impression-this is a pilot study which explores the feasibility (but not effectiveness/efficacy) and the potential messaging content to progress to a clinical trial (intent as stated in the manuscript). I do believe the findings are an important guide to developing/optimizing existing DAT. If published, it should be significantly shortened to capture description of major design elements and thematic findings.

Additional comments:

Unclear if SMS messaging directed at the patient or the parent/guardian, and how that may differ among those of different age groups. Uncertain if adherence reports directed to the patient or the healthcare workers.

Methods (p6) Explicit statement of primary and secondary objectives would greatly facilitate understanding/evaluation of methods and statistical hypothesis testing (including sample size estimates) employed for select endpoints.

Study population-Eligibility of ART for 6mo-same or variable regimens can impact compliance. Also, children (0-14yr) responses from caregivers (not patients) while adolescents responded independent of caregiver intervention, so source of perception varied between the two groups.

Survey (p7)-method of determination of adherence levels not stated.

Sampling for DAT: method to assure diversity in three variables (age, adherence levels and facilities) not described. (independently or in combination).

DAT intervention: description of the method and its functionality provided after the method of its use was provided. Method of determining adherence levels not addressed. Intervention as described includes DAT + counseling. Unclear how the behavioral model was different between children and adolescents.

Exit interviews-unclear how questionnaire on adherence was used adjunctively or in place of DAT-collected data. Unclear how responses quantified/characterized by interviewers.

Tables. Include definitions for abbreviations (such as IQR).

Table 1. HIV disclosure and sharing of phone data do not account for the total population in that category

Table 2. Total participants in each category not described. Graphical feedback method described in table not described in methods. “Network” capitalized

Affective attitude-discussion regarding “helpful tool” should be moved to discussion section. Uncertain how many participants attended boarding school (not described). Uncertain how selected quotes were chosen or whether the intent is to communicate usual or unusual observations.

Reviewer #2: Overall : The authors have attempted to provide deeper understanding the needs and requirements of SMS based medication adherence reminders in Children and adolescents living with HIV. Digital adherence tools and utility of text based reminders are required for viral suppression, the authors have attempted to share the learnings of what is working and not.

Major Issues:

The manuscript is very lengthy and lack cohesiveness and clarity. The quantitative and qualitative components needs to be appropriately explained and currently it needs additional clarity.

1. Abstract: the methods needs to be clear and succinct. Currently, it is confusing to comprehend the methodology and steps.

2. Abstract: the results stating 93% of adolescents and 83% of children's parents were interested in receiving reminders. What is the total N or denominator 284 or 40 who were in intervention?

3. Abstract: The percentages does not mention n and higher values can be interpreted differently. The results needs to be updated with apt numbers which adds more novelty and effectiveness for the readers.

4. Abstract: Language like " Adherence reports inspired good adherence" is difficult to comprehend

5. Abstract: Conclusion statement "We will implement this in a clinical trial to assess effectiveness in improving adherence" needs to be edited and the studies take home message should be highlighted in summary line.

Methodology:

The methods is very long and confusing. Not sure what is the steps or operation flow in the assessment of DAT. There is a lot of surveys, and interviews but not clear why it was chosen or performed to understand the DAT effectiveness. The methods needs to written in a comprehendible way for authors to follow the study logic and feasibility. Need a workflow consort or a diagram explaining the processes and steps taken to facilitate user understanding. The intervention sampling states participants were purposively selected, does it mean they were selected for any given reason? Then is there a selection bias?

Results:

The education back ground of the participants. Not sure the reason or impact of children education in DAT intervention, as the caregiver is has the access and control of the medications. 

Need more clarification of self reported adherence versus Pharmacy refill and DAT reported adherence metrics. How the score has been calculated. Seems from the qualitative IDI's the participants are reminded about the delay in taking the medication on time. Is the score updated when the participants take the medication after the text message. Appendix 1: The <18 population has some 2 year and 4 year old where adherence is 100% across the table and for some DAT adherence shows variations. Not clear about the interpretation.

The qualitative section of the paper has the most information about the device and text message. The manuscript should focus on this section as it provides novel information about what is working and what is not. Especially DAT interventions and feasibility shows the value of DATs to improve medication adherence. The results section needs attention to make a cohesive and well explained study findings.

Discussion: the section is very brief and requires additional comparison for the qualitative themes and quantitative findings. 

Overall the manuscript required a thorough revision and changes to make it brief and well articulation. The sample size of 40 and generating descriptive values of high percentages needs to be addressed.

6. PLOS authors have the option to publish the peer review history of their article (what does this mean?). If published, this will include your full peer review and any attached files.

**Do you want your identity to be public for this peer review?** For information about this choice, including consent withdrawal, please see our Privacy Policy.

Reviewer #1: No

Reviewer #2: No

---

## [Decision Letter · Decision Letter 1]

26 Aug 2023

Developing contents for a digital adherence tool: a formative mixed-methods study among children and adolescents living with HIV in Tanzania

PDIG-D-23-00097R1

Dear DR Swai,

We are pleased to inform you that your manuscript 'Developing contents for a digital adherence tool: a formative mixed-methods study among children and adolescents living with HIV in Tanzania' has been provisionally accepted for publication in PLOS Digital Health.

Best regards,

Haleh Ayatollahi

Section Editor

PLOS Digital Health

Reviewer Comments (if any, and for reference):

Reviewer's Responses to Questions

**Comments to the Author**

1. If the authors have adequately addressed your comments raised in a previous round of review and you feel that this manuscript is now acceptable for publication, you may indicate that here to bypass the “Comments to the Author” section, enter your conflict of interest statement in the “Confidential to Editor” section, and submit your "Accept" recommendation.

Reviewer #1: All comments have been addressed

Reviewer #2: All comments have been addressed

2. Does this manuscript meet PLOS Digital Health’s publication criteria? Is the manuscript technically sound, and do the data support the conclusions? The manuscript must describe methodologically and ethically rigorous research with conclusions that are appropriately drawn based on the data presented.

Reviewer #1: Yes

Reviewer #2: Yes

3. Has the statistical analysis been performed appropriately and rigorously?

Reviewer #1: Yes

Reviewer #2: I don't know

4. Have the authors made all data underlying the findings in their manuscript fully available (please refer to the Data Availability Statement at the start of the manuscript PDF file)?

Reviewer #1: Yes

Reviewer #2: Yes

5. Is the manuscript presented in an intelligible fashion and written in standard English?

Reviewer #1: Yes

Reviewer #2: Yes

6. Review Comments to the Author

Reviewer #1: Authors have addressed major comments/issues. Revision much improved.

Reviewer #2: The authors have revised the manuscript and addressed all the reviewers comments. There is a significant change in the manuscript structure and content has been improvised a lot. The article is well written and recommend to proceed further in the process of publication.

7. PLOS authors have the option to publish the peer review history of their article (what does this mean?). If published, this will include your full peer review and any attached files.

**Do you want your identity to be public for this peer review?** For information about this choice, including consent withdrawal, please see our Privacy Policy.

Reviewer #1: No

Reviewer #2: **Yes: **Venkatraghavan Sundaram
